# Genome-Wide Identification of the OPT Gene Family and Screening of Sb-Responsive Genes in *Brassica juncea*

**DOI:** 10.3390/plants14213399

**Published:** 2025-11-06

**Authors:** Xianjun Liu, Mingzhe Chen, Yuhui Yuan, Jialin Sheng, Pintian Zhong, Sha Gong, Zhongsong Liu, Guohong Xiang, Junhe Hu, Mingli Yan, Yong Chen, Liang You

**Affiliations:** 1College of Agriculture and Biology, Hunan University of Humanities, Science and Technology, Key Laboratory of Development and Utilization and Quality and Safety Control of Characteristic Agricultural Resources in Central Hunan of College of Hunan Province, Loudi 417000, China; 2Loudi Institute of Agricultural Sciences, Loudi 417000, China; 3Yuelushan Laboratory, Changsha 410128, China; 4College of Agronomy, Hunan Agricultural University, Changsha 410128, China; 5Crop Research Institute, Hunan Academy of Agricultural Sciences, Changsha 410125, China

**Keywords:** *Brassica juncea*, antimony, *BjOPT* genes, transcriptome sequencing, Sb-NA/PS

## Abstract

Antimony (Sb), a toxic metalloid, inhibits plant growth and threatens human health. Yellow Stripe-Like (YSL) proteins play crucial roles in metal ion transport and cellular homeostasis. While the OPT gene family has been characterized in some species, its genome-wide organization and functional involvement in Sb stress response remain unexplored in *Brassica juncea*. Here, we identified 47 high-confidence *BjOPT* genes and combined transcriptomic approaches to elucidate their regulatory roles under Sb stress. Phylogenetic tree, conserved motifs, and gene structure analyses consistently distinguished the BjOPT and BjYSL subfamilies. Comparative and collinearity analyses indicated that *OPT* genes in *Brassica* species (including *B. rapa*, *B. nigra*, and *B. juncea*) expanded independently of whole-genome triplication events. Transcriptomic profiling revealed significant enrichment of differentially expressed genes (DEGs) related to key biological processes (oxidative and toxic stress response, metal ion transport, and auxin efflux) and pathways (glutathione metabolism, MAPK signaling, and phytohormone transduction), highlighting their roles in Sb detoxification and tolerance. Notably, three *BjYSL3* (*BjA10.YSL3*, *BjB02.YSL3*, and *BjB05.YSL3*) genes exhibited strong up-regulation under Sb stress. Heterologous expression in yeast demonstrated that both *BjA10.YSL3* and *BjB02.YSL3* enhance Sb tolerance, suggesting their potential role in transporting Sb–nicotianamine (NA) or phytosiderophore (PS) complexes. These findings advance our understanding of Sb tolerance mechanisms and provide a basis for developing metal-resistant crops and phytoremediation strategies.

## 1. Introduction

China currently dominates global antimony (Sb) reserves and production, with an output of approximately 60,000 metric tons in 2024—3.53 times that of Tajikistan, the second-largest producer [1]. Severe Sb contamination has been reported in soils surrounding major mining areas in Guangxi, Hunan, Yunnan, and Guizhou provinces [2]. Sb, a potentially toxic metalloid, is listed as a priority pollutant by the U.S. Environmental Protection Agency and the European Union [3]. However, owing to its advantageous properties, Sb is extensively used in industrial and military applications, including flame retardants, batteries, alloys, glass ceramics, and catalysts [4,5]. These Sb-containing products release the metalloid into ecosystems through various pathways, disrupting ecological balance and bioaccumulating through food chains, ultimately posing risks to human health [6,7,8].

During metal uptake and translocation, plants rely on various transporter proteins localized to the plasma membrane and organellar membranes, such as vacuoles, mitochondria, and plastids [9,10]. Key transporters include the oligopeptide transporter (OPT) family, metal tolerance proteins (MTPs), ATP-binding cassette transporters (ABCs), heavy metal ATPases (HMAs), and natural resistance-associated macrophage proteins (NRAMPs) [9]. These proteins mediate the translocation of metal ions between the cytoplasm and organelles, facilitating sequestration and detoxification [10].

The OPT family, comprising the peptide transporter (PT) and YSL subfamilies, functions as proton-coupled symporters [11,12,13]. *OPT* genes were first identified in *Candida albicans* [14] and subsequently cloned in bacteria and plants. They have been demonstrated to participate in iron transport in both plants and fungi, as well as in the translocation of small peptides, glutathione, and metal chelates [15]. In *Arabidopsis*, AtOPT3 functions as a phloem-specific transporter essential for iron allocation [16]. Similarly, in rice, OsOPT1, OsOPT4 and OsOPT7 have been confirmed to transport iron chelates [17]. The YSL subfamily primarily facilitates the long-distance transport of metals, such as manganese (Mn), zinc (Zn), copper (Cu), cadmium (Cd), and nickel (Ni), chelated with nicotianamine (NA) or phytosiderophores (PSs) [15,18,19]. Research has demonstrated that in *Arabidopsis*, AtYSL1, AtYSL2, and AtYSL3, localized in leaf parenchyma cells, mediate Fe-PS translocation to stems [20,21]. In *Brassica juncea* (*B. juncea*), the plasma membrane-localized metal–NA transporter BjYSL7 facilitates Fe(II)-NA transport and contributes to Cd and Ni root-to-shoot translocation [22]. Notably, recent studies show that CRISPR-edited *OsYSL15* mutants in rice exhibit a 40.7–70.6% reduction in chromium (Cr) uptake, identifying *OsYSL15* as a key gene for Cr accumulation [18].

*B. juncea* (2*n* = 36, AABB), an economically important allopolyploid vegetable crop derived from the hybridization between *B. rapa* (*n *= 10, AA) and *B. nigra* (*n *= 8, BB), is classified into four major types based on morphological traits and agricultural use: seed mustard, leaf mustard, stem mustard, and root mustard [23]. Furthermore, it exhibits rapid growth, high biomass yield, and notable capacity for heavy metal accumulation and translocation, making it an ideal candidate for phytoremediation [24]. Advances in plant genomics have enabled comprehensive characterization of the OPT gene family in species including *A. thaliana*, rice, potato [9], tomato and maize [25]. In *B. juncea*, 27 *BjYSL* genes homologous to *AtYSL2/5/6/8* were previously identified using ARDRA-based molecular fingerprinting with degenerate primer-mediated RT-PCR [26]. Despite these advances, a systematic annotation of *BjOPT* genes remains incomplete. This study employs HMMER-based profiling to identify high-confidence OPT proteins integrated with transcriptomic analyses to reveal key *OPT* genes involved in Sb stress response. Our study aims to elucidate the regulatory mechanisms governing Sb tolerance and accumulation, thereby providing insights into plant adaptation to heavy metal stress and informing strategic development of phytoremediation approaches.

## 2. Results

### 2.1. Genome-Wide Identification and Characterization of OPTs

A genome-wide identification of OPT proteins was conducted across three *Brassica* species using the oligopeptide transporter domain (PF03169) as a query to screen the PFAM, SMART, and NCBI-CDD databases. A total of 98 *OPT* genes were identified: 47 in *B. juncea*, 24 in *B. rapa*, and 27 in *B. nigra* (Appendix A). Notably, the number of *OPT* genes in the allotetraploid *B. juncea* was slightly lower than the combined total of its diploid progenitors (*B. rapa* and *B. nigra*). All identified genes were systematically named according to chromosomal location, species abbreviation, and subfamily (e.g., *BjA02.OPT1* and *BjB02.YSL1* for *B. juncea*).

Physicochemical analysis indicated that most BjOPT proteins range from 632 to 1243 amino acids (70.83–139.25 kDa), with exceptions including the unusually long BjA04.YSL4 (3545 aa), possibly due to domain expansion, and the shorter BjA03.OPT8b (431 aa) and BjB08.OPT1 (332 aa). Theoretical isoelectric points (pI) varied from 5.65 to 9.65, and grand average of hydropathicity (GRAVY) values ranged between 0.04 and 0.65, consistent with their classification as hydrophobic transmembrane transporters. Subcellular localization predictions indicated that BjA04.YSL4 localizes to mitochondria and the nucleus, BjYSL3 (BjA10.YSL3/BjB02.YSL3/BjB05.YSL3) to the endoplasmic reticulum, and the majority of BjOPT proteins to the plasma membrane. Transmembrane helix analysis revealed 11–16 helices in most proteins, except for BjA03.OPT8b (8 helices) and BjB08.OPT1 (7 helices), corresponding to their reduced lengths. Complete physicochemical properties are provided in Appendix A.

### 2.2. Phylogenetic Analysis and Chromosomal Localization of BjOPTs

To elucidate phylogenetic relationships among OPT proteins in *Brassica* species, we conducted a comprehensive phylogenetic analysis of 115 OPT sequences from *A. thaliana, B. rapa*, *B. nigra*, and *B. juncea* (Figure 1). The neighbor-joining tree revealed two well-supported clades corresponding to the OPT and YSL subfamilies, with most nodes showing strong bootstrap support (≥90%). Among the 17 color-coded clades (based on AtOPT/AtYSL orthologs), most *B. juncea* (AABB) *OPT* gene copies corresponded to orthologs from *B. rapa* (AA) and *B. nigra* (BB). Furthermore, most branches contained one *OPT* ortholog in the diploid progenitor species and two or more in the allotetraploid, consistent with chromosomal additivity, as exemplified by *BrA09.YSL6* + *BnB07.YSL6* = *BjA09.YSL6 & BjB07.YSL6*. Notably, *BjB05.OPT6* had no detectable ortholog in the progenitors, while *BjB07.YSL5* corresponded to two copies in *B. nigra* (*BnB07.YSL5a/b*). These results indicate that the OPT gene family was largely conserved during allopolyploidization, although certain members experienced expansion or contraction.

Chromosomal localization analysis indicated that the 47 *BjOPT* genes are unevenly distributed across 14 chromosomes of *B. juncea*, with 21 and 26 genes located in the A and B subgenomes, respectively. Chromosomes B02 and B05 showed the highest gene density (6 genes each), whereas A04, A06, and B01 contained only one *BjOPT* gene each. No *BjOPT* genes were detected on chromosomes A05, A07, or B06 (Appendix A).

### 2.3. Structural Analysis of BjOPT Proteins

The functional diversity of gene families is primarily determined by variations in gene structure and conserved domain composition [27]. Analysis of 10 conserved motifs in BjOPT proteins revealed distinct subfamily-specific patterns (Appendix A). BjOPT members contained 5–10 motifs, with BjB08.OPT1 exhibiting the minimal set (motifs 1, 3, 5, 7, and 9), suggesting these constitute core functional elements. In contrast, BjYSL proteins contained only 3–4 motifs (primarily motifs 1, 2, and 8), indicating potential functional divergence between the OPT and YSL subfamilies. Domain analysis confirmed that all BjOPT proteins retain the conserved OPT domain, supporting their conserved role in substrate transport. Notably, BjA04.YSL4, the longest and most structurally complex protein, uniquely contained PKc_like and FAT domains, consistent with domain expansion (Appendix A), though the functional significance of these domains remains unclear. Gene structure analysis indicated considerable variability in CDS/UTR organization among *BjOPT* and *BjYSL* genes. For example, *BjA04.YSL4* contained 28 CDS regions, whereas *BjB08.OPT1* had only one (Appendix A), demonstrating substantial structural variation. Despite these differences, all members retained core functional domains, highlighting evolutionary conservation of transport function amid structural diversification.

### 2.4. Synteny and Divergence Time Analysis of OPT Genes

Gene duplication events serve as a major driver of family evolution [28]. The present study indicates that the *BjOPT* gene family in *B. juncea* has undergone both expansion and contraction throughout its evolutionary history. To investigate selective pressures on paralog *OPT* gene pairs within the *B. juncea* genome and between *B. juncea* and its diploid progenitors, synteny analysis was conducted among *B. juncea *&* B. juncea*, *B. juncea *&* B. rapa*, and *B. juncea *&* B. nigra*, identifying 192 orthologous gene pairs. The Ka/Ks ratios were calculated for three comparison sets (51 pairs in *B. juncea *&* B. juncea*, 63 in *B. juncea *& *B. rapa*, and 78 in *B. juncea *&* B. nigra*), all of which exhibited Ka/Ks < 1 (Figure 2b; Appendix A), indicating pervasive purifying selection (Figure 2b). Further analysis revealed that among the 47 *BjOPT* genes, there were 3 tandem duplications, 16 segmental duplications, and 32 whole-genome duplication events (Appendix A). Among these, the *BjOPT8* gene exhibited the highest number of homologous gene pairs (13 pairs), whereas only one homologous pair was identified for the BjOPT gene. The divergence time (T = Ks/2λ) was estimated for 51 duplicated gene pairs in *B. juncea*, ranging from approximately 74.0 to 4.5 million years ago (MYA), with a mean of 25.1 MYA. Three gene pairs—*BjA10.OPT9/BjB02.OPT9*, *BjA09.OPT7/BjB08.OPT7*, and *BjA04.YSL4/BjB01.YSL4*—exhibited the most recent duplication events.

Collectively, these results suggest that *OPT*s are highly conserved within *Brassica* species without significant functional divergence. Both intra- and interspecies duplicated *OPT* gene pairs may perform similar functions, particularly in mediating the transport of small peptides, glutathione, and metal chelates, providing critical insights for future functional predictions of *OPT* genes.

### 2.5. Analysis of BjOPT Promoter Elements, Expression Patterns, and GO Enrichment

Using PlantCARE software (http://bioinformatics.psb.ugent.be/webtools/plantcare/html/, accessed on 20 September 2025), cis-regulatory elements (CREs) in *BjOPT* promoters were predicted and categorized into four categories based on previous studies: phytohormone response, stress response, light response, and growth/development elements [29]. Overall, phytohormone-responsive elements such as ABA-responsive ABRE and MeJA-related CGTCA motif and TGACG motif, along with the anaerobic induction element ARE and light-responsive Box 4, were frequently identified among the *BjOPT* promoters (Appendix A), suggesting their potential roles in stress adaptation. Furthermore, no distinct distribution pattern of these elements was observed between the BjOPT and BjYSL subfamilies, indicating functional divergence in the regulatory domains among different *BjOPT* promoters.

To elucidate the tissue-specific expression profiles of *BjOPT* genes, RNA-seq data from various tissues of *B. juncea* cv. Sichuan Yellow were analyzed, including roots, stems, leaves, buds, siliques at 7 and 15 DAF, pods at 20 DAF, seeds, and seed coats. Results showed that *BjOPT3/7* and *BjYSL3/4/6* are broadly expressed across all tissues, with homologous genes such as *BjYSL3* (*BjA10.YSL3*, *BjB02.YSL3*, and *BjB05.YSL3*) exhibiting nearly identical expression patterns. In contrast, *BjOPT5* (*BjA01.OPT*5, *BjB02.OPT5*, and *BjB05.OPT5*) displayed marked tissue-specific expression, particularly in seed coats and siliques (Figure 3a). GO enrichment analysis indicated that the 47 *BjOPT* and *BjYSL* members are primarily associated with biological processes, including oligopeptide transport (GO:0006857), transmembrane transport (GO:0055085), and iron ion homeostasis (GO:0055072) (Figure 3b), supporting their involvement in transmembrane transport of oligopeptides and metal ions.

### 2.6. Transcriptome Profiling Under Sb Stress

To investigate the transcriptional response of *B. juncea* to Sb(III) stress, four-week-old plants were hydroponically exposed to 75 mg/L Sb(III). Following 24 h of treatment, plants exhibited observable wilting of basal leaves and root browning. Evans Blue staining revealed that Sb exposure caused a distinct color change in roots from white to dark blue, indicating severe disruption of plasma membrane integrity. Based on these phenotypic changes, RNA sequencing was performed on root tissues from three biological replicates each of control (CK_1, CK_2, CK_3) and Sb-treated (T_1, T_2, T_3) plants. A total of 140.06 million reads and 42.02 Gb of clean bases were obtained, with averages of 23.34 million reads and 7.00 Gb per sample. The Q20 and Q30 values for all samples exceeded 97.66% and 93.21%, respectively (Appendix A), indicating high sequencing quality. Compared to the control group, 20,454 differentially expressed genes (DEGs) were identified in the treated group, comprising 9871 up-regulated and 10,583 down-regulated genes, with up-regulated genes showing a more pronounced response to Sb stress (Figure 4a). GO enrichment analysis revealed that the DEGs were predominantly associated with stress responses (e.g., response to oxidative stress, defense response, glutathione transferase activity), metal ion binding (e.g., cellular response to nickel ion, metal ion transport, cadmium ion binding), auxin response (e.g., response to auxin, auxin efflux), and cell wall-related processes (Figure 4b; Appendix A). KEGG pathway analysis indicated significant enrichment of DEGs involved in glutathione metabolism, MAPK signaling pathway—plant, plant hormone signal transduction, phenylpropanoid biosynthesis, and ABC transporters (Figure 4c; Appendix A), highlighting their roles in the Sb stress response in roots.

Further analysis showed that most genes in the BjOPT family were barely expressed in root tissues. Under Sb stress, four *BjOPT7*, two *BjYSL1*, and two *BjYSL6* homologous genes were slightly down-regulated. Given the lack of evidence for OPT proteins participating in metal efflux or sequestration, this down-regulation may indicate a defensive tolerance mechanism against Sb toxicity. Based on previous findings that *OPT* overexpression enhances metal tolerance [22] and that loss-of-function mutants exhibit metal deficiency [21], the significant up-regulation of *BjA10.YSL3*, *BjB02.YSL3*, and *BjB05.YSL3* (Figure 4d) strongly suggests a critical positive regulatory role for BjYSL3 in Sb tolerance.

### 2.7. Functional Analysis of BjOPTs in Mediating Sb Tolerance

Transcriptome data revealed that several *BjOPT* genes responded significantly to Sb stress. To further evaluate their temporal expression patterns, we selected eight representative *BjOPT* genes (Appendix A), based on tissue-specific expression profiles and transcriptomic responses, for qRT-PCR analysis. Expression levels were quantified in roots of *B. juncea* after 12, 24, and 48 h of Sb treatment (Figure 5). Consistent with transcriptome data, all three *BjYSL3* homologs (*BjA10.YSL3*, *BjB02.YSL3*, and *BjB05.YSL3*) were up-regulated under Sb stress but exhibited distinct expression dynamics: *BjA10.YSL3* and *BjB02.YSL3* expression increased gradually, reaching 7.4-fold and 506.5-fold higher than the control at 48 h, respectively, while *BjB05.YSL3* peaked at 12 h and subsequently declined. *BjA08.OPT3* and *BjA01.OPT6* showed no significant change at 24 h but were significantly induced at 48 h. Additionally, *BjB05.OPT7* and *BjB07.YSL5* also exhibited Sb-responsive up-regulation. These results demonstrate that multiple *BjOPT* genes are up-regulated under Sb stress, implicating their potential roles in Sb-NA complex transport or detoxification mechanisms.

Based on the qRT-PCR results, the CDS sequences of *BjA10.YSL3* and *BjB02.YSL3* were cloned and inserted into the pYES2-NTB vector. The tolerance of yeast strains carrying these recombinant plasmids, as well as an empty vector control, was evaluated under 20 mM Sb(III) stress. Screening on SG-Ura solid medium demonstrated that heterologous expression of *BjA10.YSL3* and *BjB02.YSL3* significantly enhanced yeast Sb tolerance compared to the control (*Δycf1 *&* Δycf1 +* vector) (Figure 6a). Growth curves in liquid medium under Sb stress showed rapid increases in OD_600_ values across all treatments during the initial culture phase, stabilizing after approximately 20 h. However, the heterologous expression of *BjA10.YSL3* and *BjB02.YSL3* consistently resulted in higher OD values compared to the control throughout the experiment (Figure 6b). These findings suggest that both genes may play key roles in the transport of Sb-NA/PS complexes or Sb detoxification. However, whether they are directly involved in regulating Sb uptake and translocation requires further investigation.

## 3. Discussion

Sb poses significant health risks to humans, including chronic toxicity, carcinogenicity, and teratogenicity [30]. Our previous studies have demonstrated that *B. juncea* exhibits strong tolerance to multiple heavy metals (Mn^2+^, Fe^2+^, Zn^2+^, Cd^2+^, Sb^3+^, and Pb^2+^), with several *BjMTP* genes playing broad roles in the response to various heavy metal stresses [29]. The OPT family represents a group of proton-coupled transporters that play crucial roles in metal homeostasis, nitrogen mobilization, and sulfur distribution in plants [31]. Genome-wide identification of OPT family members has been completed in key plant species such as *A. thaliana* and rice, and the functions of some genes have been experimentally validated [15]. However, the identification of *OPT* genes in *B. juncea*, transcriptomic studies of its response to Sb stress, and functional characterization of BjOPT members under Sb exposure remain unreported. This study aims to identify *BjOPT* genes and integrate transcriptomic data from Sb-treated *B. juncea* to elucidate the key *BjOPT*s involved in the Sb stress response.

### 3.1. Expansion Characteristics of BjOPTs

In this study, a total of 47 BjOPTs (29 BjOPT and 18 BjYSL) were identified in the *B. juncea* genome through HMMER scanning combined with SMART, NCBI-CDD, and Pfam databases, along with 24 in *B. rapa* (13 BrOPT and 11 BrYSL) and 27 in *B. nigra* (13 BnOPT and 14 BnYSL). Notable differences in copy number were observed among *BjOPT* homologs: *BjOPT8* was present in five copies, while *BjOPT5/6* and *BjYSL2/4/6/7* each had two copies, largely consistent with the combined contributions from the two progenitor species. These findings indicate that the *OPT* genes in *Brassica* species did not undergo triplication-derived expansion [32,33] following divergence from *Arabidopsis* (which possesses 9 AtOPT and 8 AtYSL), but rather experienced localized gene expansion, as evidenced by the presence of three copies of genes such as BrYSL1 (BrA01.YSL1/BrA03.YSL1/BrA08.YSL1) in *B. rapa* and BnYSL1 (BnB02.YSL1/BnB03.YSL1/BnB05.YSL1) in *B. nigra*. Intra-genomic synteny analysis further revealed that the expansion of *BjOPT* genes occurred primarily due to whole-genome duplication (WGD) events between the A and B subgenomes, accompanied by 16 segmental duplications and 3 tandem duplications. These results align with earlier findings in turnip [34], supporting the hypothesis that segmental duplications within subgenomes represent the major mechanism for *BjOPT* expansion, while tandem duplications play a secondary role.

Furthermore, divergence time estimation of the 51 putative *BjOPT* gene pairs revealed substantial variation (average: 25.1 MYA; range: 74.0–4.5 MYA), with 34 pairs diverging between 15.3 and 4.5 MYA. Considering that the *Brassica*–*Arabidopsis* split occurred approximately 24 MYA [35], and the divergence among diploid *Brassica* species (e.g., *B. rapa*, *B. nigra*, *B. oleracea*) is estimated to be between 20 MYA [36] and 7.9 MYA [37], preceding the recent (8000–14,000 years ago) allopolyploid origin of *B. juncea* [34]. We conclude that some *BjOPT* duplications predated the *Brassica*–*Arabidopsis* divergence, while the majority occurred during the diversification of its diploid progenitor species.

### 3.2. Divergence Between BjOPT and BjYSL Subfamilies

Previous phylogenetic studies have classified OPTs into two distinct subfamilies: OPT and YSL [11]. Functional differences have also been reported: the OPT subfamily is primarily involved in the transport of small peptides, glutathione, and metal chelates, whereas YSL proteins facilitate long-distance transport of metal–NA complexes, reflecting structural and functional divergence between the two subgroups [15]. Consistent with this, our study demonstrates clear differentiation between *BjOPT* and *BjYSL* genes in the phylogenetic tree, conserved motifs, and gene structure. For instance, *BjOPTs* generally possess more conserved motifs and fewer exons compared to *BjYSL* members. Similar patterns have been observed in turnip [34] and peanut [38], suggesting evolutionary conservation of OPT and YSL subfamilies across species. Although promoter cis-element profiles and tissue-specific expression patterns did not reveal systematic differences between the two subfamilies, homologous genes within each subfamily exhibited convergent expression behavior. This aligns with earlier observations in *BjMTP* homologs [29], implying that homologous heavy metal transporters may share conserved expression characteristics and perform similar biological functions.

### 3.3. Molecular Response Characteristics of B. juncea to Sb Stress

Studies indicate that Sb induces chlorosis, impairs photosynthesis, reduces membrane stability and nutrient uptake, and promotes oxidative stress via reactive oxygen species (ROS) accumulation, thereby inhibiting plant growth and development [39]. Transcriptomic analyses have revealed transcriptional changes in response to Sb stress in many key species, including *A. thaliana* [40], *Oryza sativa* [41], *Festuca arundinacea* [42], and *B. napus* [43]. This study provides the first transcriptomic profile of the heavy metal-tolerant species *B. juncea* under Sb stress, uncovering widespread molecular changes with more than 20,000 DEGs. The abundance of DEGs may be attributed to the strong phytotoxicity of Sb, particularly its disruption of root cell membrane integrity (Appendix A). Functional enrichment analysis showed that these DEGs were significantly associated with biological processes such as response to oxidative stress, toxic substance, toxin catabolic process, metal ion transport, and auxin efflux. Consistent with reports in *F. arundinacea* under Sb stress [42], GO terms related to stress response and oxidoreductase activity were significantly enriched. KEGG analysis revealed strong enrichment in glutathione metabolism, MAPK signaling, plant hormone signal transduction, and phenylpropanoid biosynthesis. Glutathione (GSH), a key antioxidant in plants, contributes critically to ROS scavenging and detoxification under stress and acts synergistically with phytohormone signaling to improve abiotic stress tolerance [44]. The glutathione pathway is widely implicated in heavy metal detoxification [45]. MAPKs, as key signal-transducing enzymes, are commonly activated by heavy metal stress, while Cd or Cu excess in *Arabidopsis* induces NADPH oxidase, H_2_O_2_ accumulation, and MAPK activation [46]. Based on the findings of this study, the non-essential heavy metal Sb is proposed to cause irreversible phytotoxicity through a multi-step mechanism. Sb exposure initially induces oxidative stress, triggering the activation of key genes associated with glutathione metabolism, MAPK signaling, and plant hormone transduction pathways to facilitate detoxification. Simultaneously, metal ion transport DEGs are likely involved in Sb translocation and cellular compartmentalization.

### 3.4. Potential Role of BjYSL3 Genes in Sb Tolerance

YSL proteins, key members of the OPT family, are involved in the transport of metal–NA/PS complexes and the maintenance of cellular metal homeostasis [15]. Functional characterization using metal-sensitive yeast mutants has become an established method for validating gene involvement in heavy metal detoxification, uptake, or efflux [47,48,49]. In this study, transcriptomic analysis identified three homologous *BjYSL3* genes (*BjA10.YSL3*, *BjB02.YSL3*, and *BjB05.YSL3*) that were significantly responsive to Sb. Using qRT-PCR and functional complementation assays in yeast, we demonstrated that heterologous expression of *BjA10.YSL3* and *BjB02.YSL3* significantly enhanced Sb tolerance, with *BjB02.YSL3* showing more pronounced effects both in expression and complementation. *AtYSL3* in *Arabidopsis* is expressed in vascular tissues and suppressed under iron deficiency; it has been shown to transport Fe(II)-NA and Fe(III)-PS in yeast [48]. Similarly, *TcYSL3* from *Noccaea caerulescens* (formerly *Thlaspi caerulescens*) mediates Fe(II)-NA and Ni-NA transport [48], and *SnYSL3* from the hyperaccumulator *Solanum nigrum* is expressed in vascular and epidermal cells of roots and stems, transporting Fe(II)-, Cu-, Zn-, and Cd-NA complexes in yeast [47]. Consistent with these reports, the *BjYSL3* homologs identified here are broadly expressed and may play key roles in Sb-NA/PS transport and Sb stress defense. However, whether *BjYSL3* is directly involved in intracellular Sb detoxification, efflux, or reduced uptake requires further investigation.

In summary, this study systematically identifies high-confidence OPTs in *B. juncea* and elucidates their roles in Sb stress response through transcriptomics and yeast functional validation. The discovery of *BjYSL3* as a series of tolerance genes provides a key genetic basis for improving crop adaptation to metal-contaminated soils via molecular breeding strategies such as allele introgression or transgenic approaches.

## 4. Materials and Methods

### 4.1. Identification and Characterization of OPT Family Proteins

Reference sequences of *A. thaliana* OPT family members (*n *= 17) were obtained from TAIR (https://www.arabidopsis.org (accessed on 10 August 2024)). Genomic and protein sequences of *B. juncea* cv. Sichuan Yellow were retrieved from the Oilseed Molecular Breeding Database (http://www.oilseedhunan.net (accessed on 10 August 2024)). To identify OPT proteins in *B. juncea*, an HMMER search was conducted using TBtools-II (v2.028) [50] with the conserved OPT domain (Pfam: PF03169) as the query (E-value threshold < 1 × 10^−70^). Candidate proteins were further verified using SMART (http://smart.embl-heidelberg.de (accessed on 12 August 2024)), NCBI-CDD (https://www.ncbi.nlm.nih.gov/cdd (accessed on 12 August 2024)), and Pfam to ensure domain integrity. Genomic data for *B. rapa* (Chiifu v3.5) and *B. nigra* (NI100 v2) were acquired from the *Brassicaceae* database (http://brassicadb.cn/#/ (accessed on 10 August 2024)), and their OPT proteins were identified using the same workflow.

Physicochemical parameters, including molecular weight, amino acid length, isoelectric point (pI), and grand average of hydropathicity (GRAVY), were computed using TBtools-II. Subcellular localization was predicted with Plant-mPLoc (http://www.csbio.sjtu.edu.cn/bioinf/plant-multi (accessed on 18 August 2024)), and transmembrane domains were inferred using TMHMM-2.0 (https://services.healthtech.dtu.dk/services/TMHMM-2.0 (accessed on 18 August 2024)).

### 4.2. Phylogenetic, Chromosomal Location, Conserved Motif and Gene Structure Analyses

A multiple sequence alignment of OPT protein sequences from *A. thaliana*, *B. rapa*, *B. nigra*, and *B. juncea* was conducted using ClustalW in MEGA X (v11.0.13). A neighbor-joining phylogenetic tree was constructed with 1000 bootstrap replicates and visualized with iTOL (https://itol.embl.de (accessed on 20 August 2024)). Chromosomal locations of BjOPT genes were mapped based on the GFF3 annotation file of Sichuan Yellow using TBtools-II.

Conserved protein motifs were predicted using the MEME suite (http://meme-suite.org (accessed on 20 August 2024)) with the maximum number of motifs set to 10. Exon–intron structures of *BjOPT* genes were derived from the *B. juncea* genome annotation. All conserved motifs, protein domains, and gene structures were visualized using TBtools-II.

### 4.3. Collinearity and Promoter Elements Analysis of BjOPTs

Synteny analysis of OPT genes among *B. juncea*, *B. rapa*, and *B. nigra* was performed using MCScanX implemented in Tbtools-II. To evaluate selection pressures, the nonsynonymous (Ka) to synonymous (Ks) substitution ratios were calculated for syntenic gene pairs across three comparative sets: *B. juncea* vs. *B. juncea*, *B. juncea* vs. *B. rapa*, and *B. juncea* vs. *B. nigra*. The divergence time of duplication events was estimated using the formula T = Ks/(2λ), with λ set to 1.5 × 10^−8^ mutations/site/year as the divergence rate for *Brassicaceae* [34].

For cis-regulatory element analysis, a 2000 bp upstream region of each *BjOPT* gene was extracted as the putative promoter sequence. These sequences were analyzed using PlantCARE (http://bioinformatics.psb.ugent.be/webtools/plantcare/html/ (accessed on 21 August 2024)) to identify cis-elements related to plant hormones, abiotic stress, light responsiveness, and growth/development. The resulting elements were summarized and visualized through a promoter element heatmap.

### 4.4. Expression Profiling and GO Enrichment Analysis

To elucidate tissue-specific expression patterns of *BjOPT* genes, we analyzed RNA-seq datasets from various tissues of Sichuan Yellow, including roots, stems, leaves, buds, siliques at 7 and 15 days after flowering (DAF), pods at 20 DAF, seeds, and seed coats (accession numbers: SRR11787772-SRR11787783, SRR807368) from previous studies [23,29]. Gene expression levels were quantified using log2(FPKM + 1) and visualized as heatmaps. For functional annotation, all identified BjOPT family members were subjected to Gene Ontology (GO) enrichment analysis using the Omicsmart platform (https://www.omicsmart.com/RNAseq/home.html (accessed on 22 August 2024)).

### 4.5. Plant Materials and Sb Stress Treatments

Plant materials and growth conditions were consistent with previously established methods [29]. Surface-sterilized seeds of Sichuan Yellow (using 50% sodium hypochlorite) were germinated on germination beds moistened with Hoagland’s solution. After one week, seedlings were transferred to black hydroponic containers filled with 1/2 Hoagland’s nutrient solution and grown under controlled conditions: 16/8 h light/dark photoperiod, 50–60% relative humidity, and 24 ± 2 °C. Based on the previously established optimal Sb concentration for *B. napus* [43], four-week-old *B. juncea* plants were treated with 75 mg/L Sb (supplied as KSbC_4_H_4_O_7_·0.5H_2_O; Analytical Reagent; Xilong Chemical Co., Ltd., Guangzhou, China). After 24 h of treatment, fresh root tissues were stained with Evans Blue (Maokang Biotechnology Co., Ltd., Shanghai, China) (10 min), rinsed with ddH_2_O, and then imaged. Additionally, root samples from both control and treated plants were collected at 12, 24, and 48 h post-exposure, immediately frozen in liquid nitrogen, and stored at –80 °C for subsequent analysis. Three biological replicates were included for each treatment condition.

### 4.6. RNA Extraction, Transcriptome Sequencing, and Functional Annotation

Total RNA was isolated from root tissues of control and Sb-treated (12, 24, and 48 h) plants using the RNAprep Pure Plant Plus Kit (Tiangen, Beijing, China) following the manufacturer’s protocol. RNA samples from control and 24 h Sb-treated roots (*n* = 6) were submitted for RNA sequencing at Nanjing Jisi Huiyuan Biotechnology Co., Ltd. RNA quality control and library construction were performed according to a previously described method [43], followed by sequencing on an Illumina NovaSeq 2500 platform. Clean reads were aligned to the Sichuan Yellow reference genome (http://www.oilseedhunan.net/download.html (accessed on 22 April 2025)), and gene expression was quantified using FPKM.

DEGs were identified with DESeq2 using thresholds of |log_2_FC| > 1 and adjusted *p*-value < 0.05. Results were visualized via volcano plots and heatmaps. Functional enrichment analyses of DEGs were performed using the GO (https://geneontology.org/ (accessed on 27 April 2025)) and KEGG pathway (http://www.kegg.jp (accessed on 27 April 2025)) databases, with significantly enriched terms defined as those with *p* < 0.05.

### 4.7. qRT-PCR Validation and Yeast Complementation Assay

cDNA was synthesized from 1 µg of total RNA using the PrimeScript™ RT reagent kit with gDNA Eraser (Takara, Dalian, China). qRT-PCR was carried out in triplicate (biological and technical replicates) with SYBR^®^ Premix Ex Taq™ (Takara) on a CFX96 Real-Time System (Bio-Rad, Hercules, CA, USA). Gene-specific primers were designed using Primer Premier (v3.0). BjGAPDH [51] was used as the internal control, and relative expression levels were calculated using the 2^−ΔΔCT^ method.

To validate the function of key Sb-responsive *BjOPT* genes, a yeast heterologous complementation assay was performed. Gene-specific primers were designed based on the yeast expression vector pYES2-NTB (ProNet Biotech Co., Ltd., Nanjing, China; plasmid schematic provided in Appendix A), and the coding sequences of *BjOPT* genes were amplified from cDNA templates derived from *B. juncea* roots treated with Sb for 24 h. The amplified fragments were subsequently cloned into the pYES2-NTB vector using homologous recombination. The resulting recombinant plasmids were transformed into the heavy metal-sensitive yeast mutant strain *Δycf1*. Transformants were selected on SG-Ura solid medium supplemented with 20 mM, followed by incubation at 30 °C. For liquid culture assays, both control and experimental groups were grown in SG-Ura liquid medium to an OD_600_ of 0.2, followed by addition of solid KSbC_4_H_4_O_7_·0.5H_2_O to adjust the Sb concentration to 20 mM. All cultures were incubated at 30 °C with shaking, and OD_600_ values were recorded every 5 h using an SP754 UV spectrophotometer (Spectrum Instruments Co., Ltd., Shanghai, China).

## 5. Conclusions

A total of 47 BjOPTs were identified in *B. juncea* and classified into OPT and YSL subfamilies. Phylogenetic relationships, conserved motifs, and gene structures revealed significant divergence between the subfamilies, whereas promoter cis-element distribution and tissue-specific expression patterns showed no systematic divergence between these two subgroups. Comparative and collinearity analyses among *A. thaliana*, allopolyploid *B. juncea*, and its diploid progenitor species indicated that *OPT*s in *Brassica* species did not undergo triplication during WGD; instead, most *BjOPT* genes expanded primarily through WGD and segmental duplication events. Transcriptomic profiling under Sb stress uncovered substantial DEGs associated with response to oxidative stress, toxic substances, metal ion transport, and auxin efflux, as well as key pathways such as glutathione metabolism, MAPK signaling, and plant hormone signal transduction, indicating their central roles in Sb detoxification and tolerance. Furthermore, three *BjYSL3* homologs were strongly up-regulated under Sb stress. Yeast heterologous complementation assays demonstrated that *BjA10.YSL3* and *BjB02.YSL3* confer enhanced tolerance to Sb. These findings confirm that the *BjYSL3* genes in *B. juncea* play a role in Sb tolerance, laying a foundation for further investigation into its underlying mechanism.

## Figures and Tables

**Figure 1 plants-14-03399-f001:**
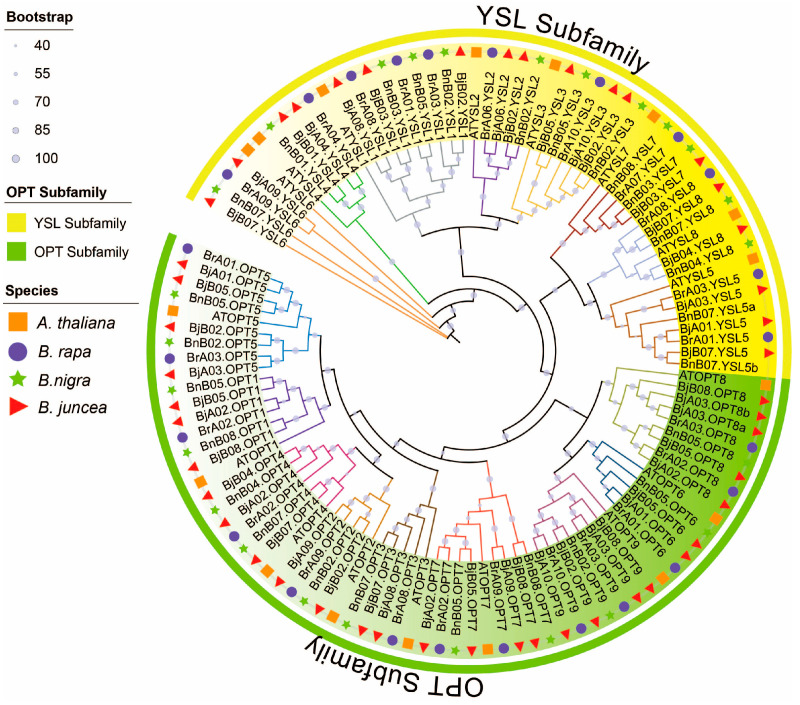
Phylogenetic relationships of OPT family proteins across *Brassicaceae* species. The neighbor-joining tree illustrates the evolutionary relationships among 115 OPT proteins from *A. thaliana* (orange squares), *B. rapa* (purple circles), *B. nigra* (green stars), and *B. juncea* (red triangles). The phylogenetic topology reveals 17 distinct clades (color-coded) representing evolutionarily conserved subgroups within the OPT family.

**Figure 2 plants-14-03399-f002:**
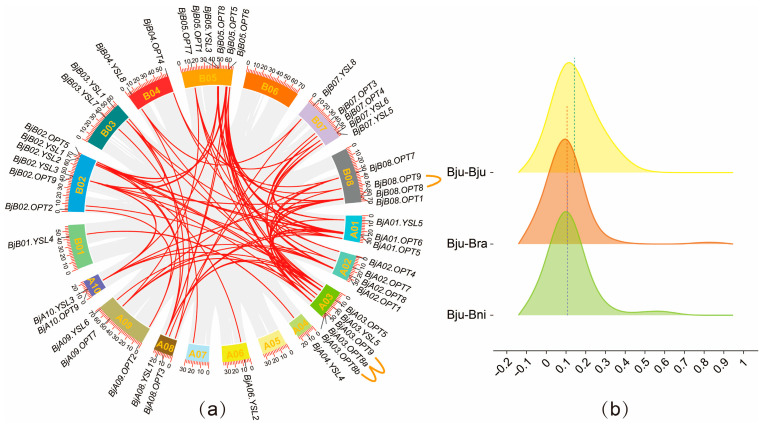
OPT gene collinearity and Ka/Ks raincloud plot analysis. (**a**) Intra-genomic collinearity analysis of *BjOPT* genes in *B. juncea*. Red lines indicate duplicated *BjOPT* gene pairs; orange lines represent tandemly duplicated pairs. (**b**) Raincloud plot of Ka/Ks ratios for orthologous *OPT* gene pairs between *B. juncea* and its diploid progenitor species.

**Figure 3 plants-14-03399-f003:**
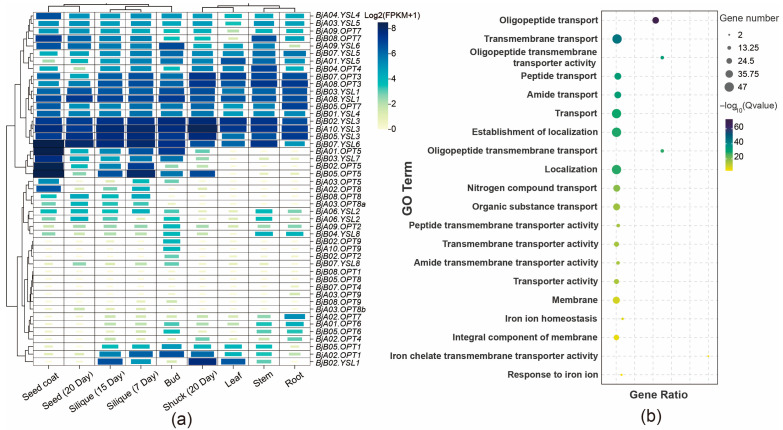
Gene expression pattern and GO enrichment analysis. (**a**) Analysis of expression pattern of *BjOPTs*. (**b**) GO enrichment analysis of *BjOPTs*.

**Figure 4 plants-14-03399-f004:**
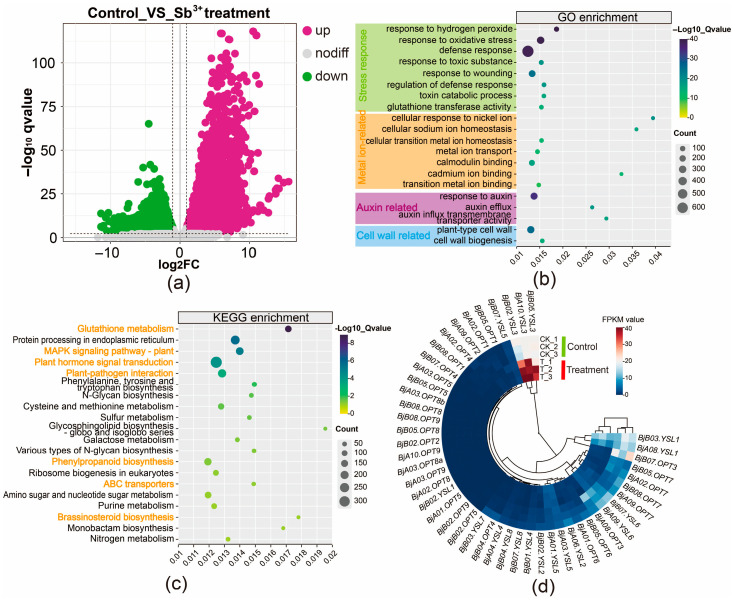
Transcriptomic analysis of root tissues in *B. juncea* under control and 24-hour Sb-treated conditions. (**a**) Volcano plot of DEGs. (**b**) Heatmap of *BjOPT* gene expression. (**c**) GO enrichment analysis. Orange font denotes significant GO terms. (**d**) KEGG pathway analysis.

**Figure 5 plants-14-03399-f005:**
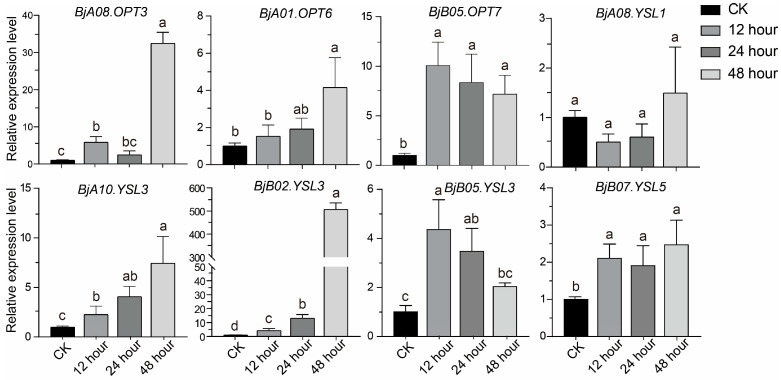
Expression patterns of eight representative genes under Sb stress at different time points. Different lowercase letters represent significant differences at *p* < 0.05 level using one-way ANOVA and Tukey’s correction.

**Figure 6 plants-14-03399-f006:**
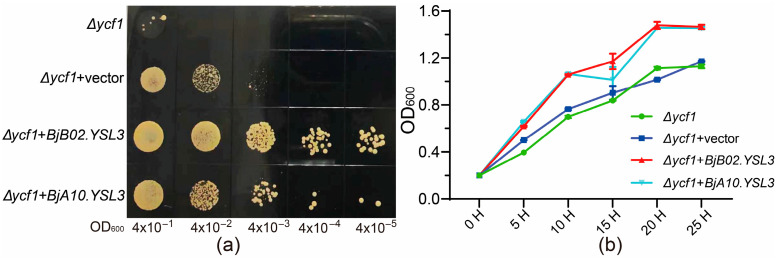
Analysis of heterologous gene expression under 20 mM Sb stress. (**a**) Screening of transformants with different OD values on SG-Ura solid medium. The labels from left to right indicate the OD_600_ values of serially diluted transformants. (**b**) OD_600_ values at different time points in SG-Ura liquid medium culture. *n* = 6.

## Data Availability

The raw sequencing data for Sb treatment and control are available under the National Genomics Data Center (NGDC) accession PRJCA046671, whereas the RNA-seq data from various *Brassica juncea* tissues are accessible under NCBI SRA accessions SRR11787772, SRR11787777, SRR11787776, SRR11787782, SRR11787779, SRR11787783, SRR11787780, SRR11787781, and SRR807368.

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
