# Peer review of "Genome-Wide Identification of the OPT Gene Family and Screening of Sb-Responsive Genes in Brassica juncea"

_plants, 2025, doi:10.3390/plants14213399_

Round 1

Reviewer 1 Report

Comments and Suggestions for Authors

The manuscript has useful result explaining the transcriptomic profiling to identify responsive genes and their validation in yeast. However, the manuscript has few issues that should be address before considering it for publication. 
The enhanced yeast growth under Sb stress is evidence for tolerance, however, phrases like "likely involved in the transport of Sb–nicotianamine (NA) or phytosiderophore (PS) complexes" (abstract, conclusions) should be amended to make the manuscript more rationale.  

I suggest the author to emphasize tolerance and suggest transport as a hypothesis for future work. 
The results of the study including the identification of ~20,000 DEGs is impressive, but author should explain for such high number. Another issue that I noticed is that the Figure 4b shows down-regulation of some OPTs so expand the discussion on their potential negative regulatory roles.  
The author should explain how the divergence times (4.5–74 MYA) align with Brassica speciation events? He can add a brief comparison with OPT evolution in other crops like rice, potato that will strengthen the study. 
The title includes "Regulatory Genes" that seems to be misleading, as OPTs are primarily transporters. I suggest to revise the title accordingly. 
 There are few issues like spacing with in many places in the manuscript so read the whole manuscript thoroughly. 
Specify the RNA-seq platform in methodology
There are some issues with the references like one I noticed is the publication status of the manuscript from 2024/2025.    
 Add a legend in Figure 1 for species symbols if present. 
 The Figure 5 have clear statistical letters but better to specify the test (ANOVA + Tukey) in the caption.
The author should expand the the practical implications of the study and explain how it can contribute in breeding programs?

Author Response

The manuscript has useful result explaining the transcriptomic profiling to identify responsive genes and their validation in yeast. However, the manuscript has few issues that should be address before considering it for publication:

Comments 1: [The enhanced yeast growth under Sb stress is evidence for tolerance, however, phrases like "likely involved in the transport of Sb-nicotianamine (NA) or phytosiderophore (PS) complexes" (abstract, conclusions) should be amended to make the manuscript more rationale. I suggest the author to emphasize tolerance and suggest transport as a hypothesis for future work.]

Response1 : [We sincerely appreciate your valuable feedback. We have carefully revised the manuscript to address the issues you raised (abstract, conclusions). As you correctly noted, our current experimental results support the conclusion that BjYSL3 enhances Sb tolerance in yeast, but do not yet provide evidence for its specific transport function.]

Comments 2:[The results of the study including the identification of ~20,000 DEGs is impressive, but author should explain for such high number. Another issue that I noticed is that the Figure 4b shows down-regulation of some OPTs so expand the discussion on their potential negative regulatory roles.]

Response 2: [(a) We fully share your interest in understanding the substantial number of DEGs induced by Sb stress. As reported in our study, Sb treatment caused rapid leaf wilting and significant root browning in B. juncea seedlings. Evans Blue staining confirmed severe damage to root cells (Line 214-218, Fig. S4). Furthermore, RNA extracted from roots after 72 h of Sb exposure showed severe degradation, making it unsuitable for RNA-seq analysis. Based on these observations, we propose that the extensive transcriptional changes likely result from the marked phytotoxicity of 75 mg/L Sb on root tissues. (b) Regarding the down-regulation of certain OPT genes, we conducted an extensive literature review and found no reports linking OPT members to metal efflux or sequestration. We therefore hypothesize that this down-regulation under Sb stress may represent a defensive tolerance mechanism, potentially reducing Sb uptake by decreasing the expression of relevant transporters (Line 241-243).]

Comments 3: [The author should explain how the divergence times (4.5-74 MYA) align with Brassica speciation events? He can add a brief comparison with OPT evolution in other crops like rice, potato that will strengthen the study.]

Response 3: [(a) We sincerely appreciate your suggestions. In the revised manuscript, we have incorporated a comparative analysis of divergence times among Brassica species, Arabidopsis, diploid Brassica species, and the allopolyploid B. juncea based on published literature. By integrating these data with divergence time estimates from the 51 homologous gene pairs identified in this study, we infer that a subset of BjOPT duplication events predate the Brassica-Arabidopsis split, while the majority occurred during the diversification of its diploid progenitor species, all preceding the formation of allopolyploid B. juncea. These results provide strong evidence that most BjOPT genes originated from the diploid parental species (Line 318-326). (b) We appreciate the suggestion to include evolutionary data from rice and potato. However, we have reservations for two reasons. First, current literature on OPT/YSL proteins focuses primarily on functional characterization, with limited studies on their evolutionary relationships. Second, our discussion centers on OPT evolution within the Brassica lineage (diploid progenitors B. rapa and B. nigra, allopolyploid B. juncea, and putative ancestor Arabidopsis). Since rice and potato data are not directly relevant to this evolutionary context, we have chosen not to include them to maintain focus. We thank you for your understanding.]

Comments 4: [The title includes "Regulatory Genes" that seems to be misleading, as OPTs are primarily transporters. I suggest to revise the title accordingly.]

Response 4: [Thank you for your suggestion. After careful consideration, we have revised the title to "Genome-Wide Identification of the OPT Gene Family and Screening of Sb-Responsive Genes in Brassica juncea" and would greatly appreciate your further feedback.]

Comments 5: [There are few issues like spacing with in many places in the manuscript so read the whole manuscript thoroughly.]

Response 5: [Thank you for your careful review and for identifying these errors. We have thoroughly reread the manuscript and made corresponding corrections. We believe the revised version has addressed all the issues you raised.]

Comments 6: [Specify the RNA-seq platform in methodology.]

Response 6: [Our RNA-seq methodology follows a protocol established in our group's recent transcriptome paper. However, based on your valuable feedback, we have added the sequencing platform details to enhance methodological clarity (Line 469-470).]

Comments 7: [There are some issues with the references like one I noticed is the publication status of the manuscript from 2024/2025.]

Response 7: [Thank you for pointing out the problems with the references. We have conducted a full review and verification of all citations in the updated manuscript.]

Comments 8: [Add a legend in Figure 1 for species symbols if present. The Figure 5 have clear statistical letters but better to specify the test (ANOVA + Tukey) in the caption.]

Response 8: [Regarding your questions about Figure 1 and Figure 5: In Figure 1, different shapes (e.g., stars, squares, triangles) and colors represent distinct species. In Figure 5, different lowercase letters indicate significant differences (p < 0.05) based on one-way ANOVA with Tukey's correction, as detailed in the figure captions (Line 263-266). We hope this clarifies your concerns.]

Comments 9: [The author should expand the the practical implications of the study and explain how it can contribute in breeding programs?]

Response 9: [As you rightly emphasized, fundamental research should ultimately serve practical applications. Accordingly, we have revised the discussion to include the potential application of Sb-responsive genes for developing metal-tolerant crops through allele introgression or transgenic breeding (Line 393-397). Prior to application, however, the functions of these genes require further validation via transgenesis or genome editing.]

Reviewer 2 Report

Comments and Suggestions for Authors

The manuscript is focusing on the study of antimony to Brassica juncea cv. Sichuan Yellow and how the presence of Sb influence the expression of OPT and YSL genes. Authors identified possible protective/changed transcription of BjYSL3 genes – mainly BjA10.YSL3, BjB02.YSL3 and BjB05.YSL3 that are probably the most responsible in semimetal-resistant of B. juncea under Sb stress due their high up-regulation. The design of experiments, novelty and preparation of this manuscript is on very high level. I found only some small type errors and necessities to adding into the text. For this reason I have following recommendations:

  • Line 134 – probably “green stars” instead of “brown stars” for nigra
  • I recommend to add at line 194 the full name for plant due Methodology is at the end of manuscript (“Brassica napus Sichuan Yellow“)
  • 3a – could be higher rows of this map to better readability of genes at the right border?
  • The full name for Sb-NA/PS complexes should be inserted also at the first mention in the text. It is only recommendation.
  • 6a – what is on this picture – Petri dishes at different time? If design is same as for Fig. 6b, could authors insert at „x-axis“ below the photos the time for dishes?
  • Latin names have to be in italic font – line 275, 281, 374.
  • Which Hoagland solution was used – 1 or ½ (line 422 and )?
  • Line 426 – please, add the purity of used antimony compound including of origin (company and country).
  • Line 453 – missing origin of vector pYES2-NTB – it was created by authors? If yes, please, insert the genes map of vector.
  • Insert the type, company and country of production of used UV spectrophotometer at line 462, please.

Author Response

The manuscript is focusing on the study of antimony to Brassica juncea cv. Sichuan Yellow and how the presence of Sb influence the expression of OPT and YSL genes. Authors identified possible protective/changed transcription of BjYSL3 genes - mainly BjA10.YSL3, BjB02.YSL3 and BjB05.YSL3 that are probably the most responsible in semimetal-resistant of B. juncea under Sb stress due their high up-regulation. The design of experiments, novelty and preparation of this manuscript is on very high level. I found only some small type errors and necessities to adding into the text. For this reason I have following recommendations:

Comments 1: [Line 134 – probably “green stars” instead of “brown stars” for nigra.] 

Response 1: [Thank you for noting the errors in the paper. We have made the necessary corrections and carried out a complete review of the entire manuscript.]

Comments 2: [I recommend to add at line 194 the full name for plant due Methodology is at the end of manuscript (“Brassica napus Sichuan Yellow“)]

Response 2: [Thank you for your constructive feedback; we have incorporated your suggestions into the revised version (Line 198).]

Comments 3: [3a – could be higher rows of this map to better readability of genes at the right border?]

Response 3: [We appreciate your valuable feedback. In response, we have modified Figure 3a by increasing the row height to prevent gene name overlap and ensure readability.]

Comments 4: [The full name for Sb-NA/PS complexes should be inserted also at the first mention in the text. It is only recommendation.]

Response 4: [We apologize for any confusion caused by the terminology. The abbreviation "Sb-NA/PS" is not a standard term; it was introduced in Line 65 as "metal-nicotianamine (NA) or phytosiderophores (PS)" and subsequently abbreviated for brevity in the text.]

Comments 5: [6a – what is on this picture – Petri dishes at different time? If design is same as for Fig. 6b, could authors insert at „x-axis“ below the photos the time for dishes?]

Response 5: [We are grateful for your attention to detail in identifying the error in Figure 6a. The figure depicts the growth response of various yeast transformants to Sb stress on solid medium at different dilution levels, with the x-axis indicating the dilution factor (decreasing from left to right). The necessary correction has been made (Line 280-283).]

Comments 6: [Latin names have to be in italic font – line 275, 281, 374.]

Response 6: [All the points you raised have been carefully addressed in the manuscript as requested.]

Comments 7: [Which Hoagland solution was used – 1 or ½ (line 422 and )?]

Response 7: [We apologize for this oversight in the original manuscript. The experiment used 1/2 Hoagland’ s nutrient solution, which has now been clearly stated in the Methods section of the revised version (Line 453).]

Comments 8: [Line 426 – please, add the purity of used antimony compound including of origin (company and country). Insert the type, company and country of production of used UV spectrophotometer at line 462.]

Response 8: [In accordance with your feedback, we have supplemented the manufacturer details and country of origin for the Sb chemical and UV spectrophotometer used in this study.]

Comments 9: [Line 453 – missing origin of vector pYES2-NTB – it was created by authors? If yes, please, insert the genes map of vector.]

Response 9: [pYES2-NTB is a yeast expression plasmid engineered by ProNet Biotech Co., Ltd. (China). Following your suggestion, we have added the plasmid map (Fig. S5) and supplier details to the manuscript (Line 487-488).]

Reviewer 3 Report

Comments and Suggestions for Authors

The manuscript presents interesting results concerning the transcriptomic analysis of Sb stress response in Brassica juncea. Particular attention was paid to OPT and YSL subfamily genes. The obtained data have a significant degree of novelty. However, a number of concerns have arisen when analyzing the manuscript:

  1. A significant shortcoming of this work is the lack of the analysis of Sb effects on morphometric parameters such as dry or fresh biomass or other growth parameters (e.g. number of leaves, root/shoot length etc.). These parameters are the integral indicator of the manifestation of the toxic effects of metals and metalloids on plants. Without such data (at least plant biomass) it is impossible to assess how toxic the used Sb concentration was. It would also be beneficial to provide some photographs of the control and Sb-treated plants. Were there any signs of leaf chlorosis or necrosis? Otherwise it is unclear how toxic the used concentration of Sb (75 mg/L Sb(III)) was. How was the Sb concentration chosen? Is important to include this into the manuscript.
  2. The Sb content in plants was not analyzed, which is also a significant drawback. The lack of a quantitative analysis of Sb accumulation in plants does not allow us to assess its entry into the plant, which is fundamentally important for explaining the observed effects and correct interpretation of the data obtained.
  3. Please provide a description of the composition of Hoagland`s solution used.
  4. The hyperaccumulator of Zn, Ni и Cd Thlaspi caerulescens currently belongs to the genus Noccaea (line 359).

Author Response

The manuscript presents interesting results concerning the transcriptomic analysis of Sb stress response in Brassica juncea. Particular attention was paid to OPT and YSL subfamily genes. The obtained data have a significant degree of novelty. However, a number of concerns have arisen when analyzing the manuscript.

Comments 1: [A significant shortcoming of this work is the lack of the analysis of Sb effects on morphometric parameters such as dry or fresh biomass or other growth parameters (e.g. number of leaves, root/shoot length etc.). These parameters are the integral indicator of the manifestation of the toxic effects of metals and metalloids on plants. Without such data (at least plant biomass) it is impossible to assess how toxic the used Sb concentration was. It would also be beneficial to provide some photographs of the control and Sb-treated plants. Were there any signs of leaf chlorosis or necrosis? Otherwise it is unclear how toxic the used concentration of Sb (75 mg/L Sb(III)) was. How was the Sb concentration chosen? Is important to include this into the manuscript. ]

Response 1: [We sincerely appreciate your constructive feedback. Your perspective highlights the importance of a comprehensive evaluation of Sb's impact on B. juncea seedlings, encompassing morphology, biomass, and other growth parameters, to fully assess its toxicity, which indeed represents a valuable research direction. However, the primary aim of this study is to first identify metal-transport related BjOPT genes at the genome-wide level in B. juncea, then treat seedlings with an optimal Sb concentration (75 mg/L) previously determined in B. napus (Line 455), and subsequently combine RNA-seq to pinpoint key BjOPT genes involved in Sb tolerance. Finally, we employed qRT-PCR and yeast heterologous expression to functionally validate the roles of these BjOPT genes in Sb tolerance.

Additionally, within the limited revision period, we have supplemented observations on the phenotypic effects of 75 mg/L Sb stress on B. juncea seedlings. The results showed noticeable wilting of basal leaves and root browning after 24 h of Sb exposure. Evans Blue staining further indicated that Sb rapidly damages root cell membranes, leading to intense blue staining of the tissues (Fig. S4). These findings, along with the detection of > 20,000 DEGs, suggest substantial root impairment from Sb toxicity.]

Comments 2: [The Sb content in plants was not analyzed, which is also a significant drawback. The lack of a quantitative analysis of Sb accumulation in plants does not allow us to assess its entry into the plant, which is fundamentally important for explaining the observed effects and correct interpretation of the data obtained.]

Response 2: [Thank you for your encouraging and insightful comments. We fully agree on the need to evaluate Sb's comprehensive effects on B. juncea, and our future work will indeed include integrated analyses of phenotype, Sb accumulation, physiological indices, and metabolomics. Nevertheless, the present study is specifically designed to screen for critical BjOPT genes underlying Sb tolerance using transcriptome data, as addressed previously. Your understanding and support of our current focus would be highly appreciated.]

Comments 3: [Please provide a description of the composition of Hoagland`s solution used.]

Response 3: [We apologize for this oversight in the original manuscript. The experiment used 1/2 Hoagland’ s nutrient solution, which has now been clearly stated in the Methods section of the revised version (Line 453).]

Comments 4: [The hyperaccumulator of Zn, Ni и Cd Thlaspi caerulescens currently belongs to the genus Noccaea (line 359).]

Response 4: [Your suggestion is much appreciated. For taxonomic accuracy, we have revised the text to Noccaea caerulescens (formerly Thlaspi caerulescens) in the updated manuscript (Lines 384-385).]

Round 2

Reviewer 1 Report

Comments and Suggestions for Authors

authors have revised thte manuscript

Author Response

Comments 1: [Authors have revised the manuscript.]

Response1: [We sincerely appreciate your positive feedback on our revisions. Your recognition serves as the greatest encouragement for the value of our research work.]

Reviewer 2 Report

Comments and Suggestions for Authors

The manuscript was sufficiently improved and I have only one note:

Line 384 – B. juncea should be written in italic font.

Author Response

Comments 1: [The manuscript was sufficiently improved and I have only one note:

Line 384 – B. juncea should be written in italic font. ]

Response1: [We sincerely appreciate your thorough review and for identifying the minor errors. There is no doubt that your suggestions will further improve the quality of our work.]

Reviewer 3 Report

Comments and Suggestions for Authors

The manuscript has been substantially revised. The figure showing the plants and root staining with Evans Blue is a good addition which helps to evaluate the physiological state of the plants at least to some degree.

Line 211-  I would not say it looks like "pronounced" leaf wilting and root browning. In this figure it looks quite moderate. (Which is good, actually. There is no point in analyzing half-dead plants). I recommend changing "pronounced" for "moderate".

Lines 234-235 - please add a reference to the corresponding figure.

Figure 4  - please indicate the duration of plant treatment with Sb in the figure caption. Was it 24h?

Author Response

Comments 1: [Line 211-  I would not say it looks like "pronounced" leaf wilting and root browning. In this figure it looks quite moderate. (Which is good, actually. There is no point in analyzing half-dead plants). I recommend changing "pronounced" for "moderate". ]

Response 1: [Thank you for this precise observation. To better reflect the phenotypic changes in B. juncea under Sb stress, we have revised the wording by replacing "pronounced" with the more appropriate term "observable" (Line 215).]

Comments 2: [Lines 234-235 - please add a reference to the corresponding figure.]

Response2: [We sincerely appreciate you bringing this issue to our attention. Initially, for layout considerations in Figure 4, we placed the heatmap (Fig. 4d) ahead of the GO (Fig. 4b) and KEGG (Fig. 4c) analyses for visual coherence. After careful reconsideration, we have revised the layout (Fig. 4, Line 235) to follow the logical sequence of content presentation, thereby improving readability for the audience.]

Comments 3: [Figure 4  - please indicate the duration of plant treatment with Sb in the figure caption. Was it 24h?]

Response3: [We apologize for the oversight in the Figure 4 caption that caused confusion. In the revised version, we have added the Sb treatment duration (24 hours) to improve clarity for readers (Line 236).]